# Geometry-Guided Conditional Adaption for Surrogate Models of Large-Scale 3D PDEs on Arbitrary Geometries

## Abstract

Deep learning surrogate models aim to accelerate the solving of partial differential equations (PDEs) and have achieved certain promising results. Although several main-stream models through neural operator learning have been applied to delve into PDEs on varying geometries, they were designed to map the complex geometry to a latent uniform grid, which is still challenging to learn by the networks with general architectures. In this work, we rethink the critical factors of PDE solutions and propose a novel model-agnostic framework, called 3D Geometry-Guided Conditional Adaption (3D-GeoCA), for solving PDEs on arbitrary 3D geometries. Starting with a 3D point cloud geometry encoder, 3D-GeoCA can extract the essential and robust representations of any kind of geometric shapes, which conditionally guides the adaption of hidden features in the surrogate model. We conduct experiments on the public Shape-Net Car computational fluid dynamics dataset using several surrogate models as the backbones with various point cloud geometry encoders to simulate corresponding large-scale Reynolds Average Navier-Stokes equations. Equipped with 3D-GeoCA, these backbone models can reduce their L-2 errors by a large margin. Moreover, this 3D-GeoCA is model-agnostic so that it can be applied to any surrogate model. Our experimental results further show that its overall performance is positively correlated to the power of the applied backbone model.

## 1 Introduction

The Partial differential equation (PDE) is a powerful model to describe various physical phenomena and help us to understand this world to a large extent. However, most PDEs do not have closed-form solutions, which leads to a resort to numerical methods for solving them. Actually, various approaches have been proposed, including finite difference (Strikwerda, 2004) and finite element methods (Hughes, 2012), whereas these methods usually have high computational costs, which are unendurable in many real-time settings. As a data-driven method, the deep learning surrogate model can learn from numerical solutions to a family of PDEs and generalize well to the unseen equations via forward propagation within a second, which is much faster than traditional numerical solvers, exhibiting a promising future.

Most traditional numerical solvers simulate PDEs with varying geometries on irregular mesh grids. Although one can form the input to uniform grids and then adopt convolution-based architectures to train the surrogate model, such as U-Net (Ronneberger et al., 2015), this process is less efficient and might introduce extra interpolation error (Li et al., 2022). Therefore, several researchers adopted Graph Neural Networks (GNN) as the backbone of surrogate model (Belbute-Peres et al., 2020; Pfaff et al., 2020). Moreover, Bonnet et al. (2022b;a) proposed benchmarking graph-mesh datasets taking graph or point cloud as the input data. In their work, and Point Cloud Networks were used to study 2D steady-state incompressible Navier-Stokes equations.

Another mainstream of the research falls into the Neural Operator Learning paradigm, whose target is to learn a mapping between infinite-dimensional function spaces. Li et al. (2020b) made certain theoretical analyses and proposed a novel iterative architecture using the kernel integral operator. Fast Fourier Transform (FFT) was applied to implement the kernel integral operator when the input

type was uniform grids. This operator, known as FNO (Li et al., 2020b), transforms features in the physical and spectral domain. Graph Neural Operator (GNO) (Anandkumar et al., 2020) was proposed to handle irregular grid input, where kernel integral operator was formulated as a message passing on a radius graph. These two neural operators have been popular and have been developed into many other improved methods capable of solving PDEs with varying geometries, such as Geo-FNO (Li et al., 2022), Factorized FNO (F-FNO) (Tran et al., 2022), and Multipole GNO (MGNO) (Li et al., 2020a).

While the work above has achieved remarkable progress in solving 2D equations, many real-world applications face the problem of 3D PDEs on varying geometries, ranging from industrial and engineering design to real-time physics simulation engines in games and virtual reality. However, when it comes to solving more complex 3D problems, the mentioned state-of-the-art approaches have severe limitations as follows:

**Inefficient Representation for 3D Inputs:** Most existing approaches treat each position in the field equally and coarsely feeds all grids into the model (Bonnet et al., 2022a;b). However, this may increase the difficulty to learn geometry features of the problem domain, since the boundary points is actually much more informative than other points for PDE solving. In this sense, existing approaches inefficiently represent the input field. This limitation becomes even more significant as the input dimension increases. To our knowledge, very few works have addressed this issue. The only previous work we found is Geo-FNO (Li et al., 2022), which suggests learning a geometric deformation between the computational domain to latent uniform grids and then feeding the data into an FNO-based architecture. However, learning an accurate coordinate transformation is quite difficult, and in some settings, the problem domain may not be diffeomorphic to any uniform grids. Concurrent work (Li et al., 2023b) have further revealed that such a strategy of geometric mapping does not help improve the overall performance. This raises an essential question of how to efficiently and effectively fuse geometry information into the field representation.

**Poor Generalization on Limited Training Samples:** Another limitation lies in data scarcity. In the field of deep learning surrogate models, generating a dataset is usually computationally exhaustive and time-consuming. For instance, creating a Computational Fluid Dynamics (CFD) dataset containing 551 samples, (Li et al., 2023b) ran large-scale 3D simulations on 2 NVIDIA V100 GPUs and 16 CPU cores, with each one taking 7 to 19 hours to complete. The small number of training samples further increases the difficulty of learning geometry features generalizable to unknown shapes.

To overcome the above challenges, we propose a brand new model-agnostic framework, 3D Geometry-Guided Conditional Adaption (3D-GeoCA). Based on a general deep learning architecture, 3D-GeoCA adopts a novel method that conditionally guides the adaption of hidden features with latent geometry representations. The involved point cloud geometry encoder has low computational costs since boundary points occupy a very small portion of the input field. Regarding the problem of data scarcity, we apply weight transfer, utilizing pre-trained point cloud models. Equipped with 3D-GeoCA, the backbone model becomes more geometry-aware and generalizes better on small-sample 3D PDE datasets. The main contributions of our paper are as follows:

1. We propose a novel framework, called 3D Geometry-Guided Conditional Adaption (3D-GeoCA), for solving large-scale 3D PDEs on arbitrary geometries. 3D-GeoCA originally introduces a point cloud geometry encoder to encode the boundary of the problem domain, and conditionally guides the adaption of hidden features in the backbone model with geometry information. Experimental results demonstrate that our framework provides generalizable geometry features beneficial to the backbone surrogate model, which is lacking in other approaches.

2. Our 3D-GeoCA is model-agnostic and orthogonal to various deep learning based 3D PDE frameworks, including MLP, GNN, GNO and so on.

3. To the best of our knowledge, our framework unprecedentedly introduces 3D understanding pre-training to the deep surrogate model for PDEs to alleviate the shortage of training samples, bridging the relationship between these two fields.

## 2 PROBLEM SETTING AND PRELIMINARIES

**Problem setting**. We consider a family of PDEs with varying domains of the following general form:

$$
\begin{aligned}
\frac{\partial u(x,t)}{\partial t} &= \mathcal{L}_a u(x,t), \qquad (x,t) \in D_\omega \times T \\
u(x,0) &= f(x), \qquad x \in D_\omega \\
u(x,t) &= g(x,t), \qquad x \in \partial D_\omega \times T,
\end{aligned}
\tag{1}
$$

where $\mathcal{L}_a$ is a differential operator describing the governing equation and is parameterized by $a$; $f$ and $g$ denote corresponding initial and boundary condition; and $D_\omega$ is the problem domain, parameterized by some latent parameters $\omega \in \Omega$.

In practical applications, we ideally assume that there exists a map $\mathcal{F} : (a, f, g, D_\omega) \mapsto u$ that gives the solution of equations 1. When we consider the steady-state equations where $u$ is independent of the time $t$, equations 1 convert to $\mathcal{L}_a u(x) = 0$ and the solution map simplifies to $\mathcal{F} : (a, g, D_\omega) \mapsto u$, from which we clearly aware that the boundary of the domain, $\partial D_\omega$, is a decisive factor to the solution $u$.

However, learning the geometry of $\partial D_\omega$ from a small dataset is challenging, especially for 3D cases. We believe this is one of the bottlenecks current studies have to confront. Considering that the boundary $\partial D_\omega$ can be discretized to the point cloud data, we introduce a point cloud encoder to enrich the learning of geometries. Moreover, a state-of-the-art 3D understanding pre-training framework, ULIP-2 (Xue et al., 2023b), is adopted to strengthen our encoder. By using point cloud models pre-trained on large-scale 3D object datasets, we can learn better geometry features to solve this dilemma.

**Preliminaries: ULIP-2**. Deriving from the **U**nified Representation of **L**anguage, **I**mages, and **P**oint Clouds (ULIP) framework proposed by Xue et al. (2023a), ULIP-2 is a tri-modal pre-training framework, which leverages a pseudo self-supervised contrastive learning approach to align features across: (i) 3D shapes, (ii) their rendered 2D image counterparts, and (iii) the language descriptions of 2D images of all views. Among them, language descriptions of 2D images come from BLIP-2 (Li et al., 2023a), a large multimodal model. In ULIP-2, a fixed and pre-aligned language-vision model, SLIP (Mu et al., 2022), is used to extract text and image features, after which the authors train point cloud encoders under the guidance of 3D-to-image and 3D-to-text contrastive alignment losses. ULIP-2 yields significant improvements on downstream zero-shot and standard classification tasks, showing a powerful capability for 3D representation learning.

## 3 3D GEOMETRY-GUIDED CONDITIONAL ADAPTION

In this section, we present our framework, 3D-GeoCA, in detail. Figure 1 illustrates the main architecture of 3D-GeoCA. As a model-agnostic framework, 3D-GeoCA consists of three main components: (i) a point cloud geometry encoder, (ii) an arbitrary backbone model, and (iii) geometry-guided conditional adaptors. The point cloud encoder takes merely the boundary of the problem domain as input, extracting its geometry features, while the backbone model considers the whole problem domain, along with the signed distance function (SDF) and normal vector features of each grid. As a core part of our framework, several geometry-guided conditional adaptors are embedded in the backbone model to conditionally guide the adaption of hidden features according to different input geometries.

**Point cloud geometry encoder**. As per previous discussions in section 2, one of the bottlenecks of current work is the under-exploited geometry features of various problem domains. To overcome this difficulty, we propose a point cloud encoder $E_P$ specialized to extract features of different geometries, whose input is a 3D point cloud $\mathbb{P} = \{\boldsymbol{x}_1^{\partial D}, \boldsymbol{x}_2^{\partial D}, \cdots, \boldsymbol{x}_n^{\partial D}\} \subset \partial D$ discretized from the boundary of the problem domain $D$.

Compared to the whole problem domain $D$, the point cloud $\mathbb{P} \subset \partial D$ usually contains a tiny part of the input grids (especially for 3D settings), thus leading to a relatively low computational cost. Current work usually under-emphasizes grids in $\mathbb{P} \subset \partial D$ and coarsely feeds all grids with simple hand-crafted features (such as SDF) into their model (Bonnet et al., 2022b;a). However, this may

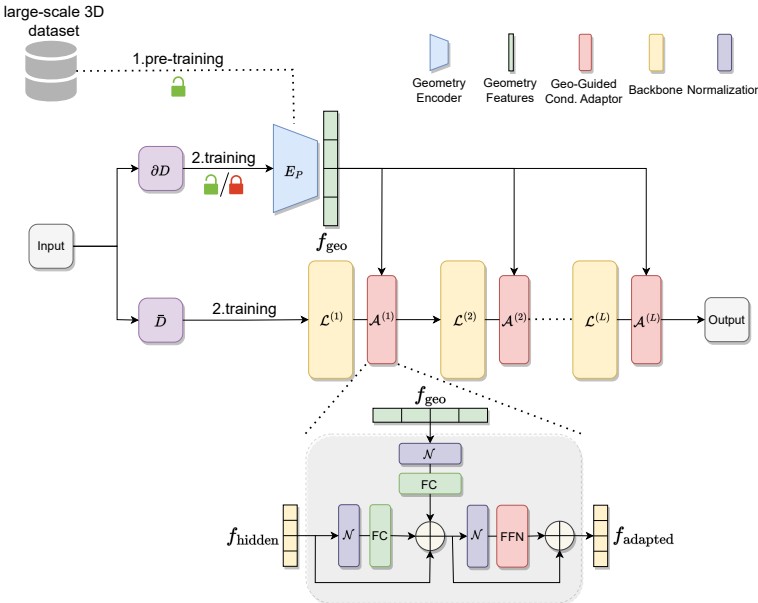

Figure 1: The main architecture of the 3D-GeoCA framework. Our 3D-GeoCA originally introduces a point cloud geometry encoder to encode the PDEs problem domain. Geometry-guided conditional adaptors after each backbone layer are designed to guide the adaption of hidden features in the surrogate model.

lead to a loss in the learning of geometries, as the small portion of grids at the boundary contains most underlying geometry information of the problem domain $D$.

To improve our encoder $E_P$, we employ a state-of-the-art 3D understanding pre-training framework, ULIP-2, to pre-train it on large-scale 3D object datasets. Once the encoder is pre-trained, the parameters of $E_P$ can either be fine-tuned or fixed. In the latter case, we can further reduce the number of trainable parameters and shorten the training process without seriously harming the effectiveness of our framework. See section 4.3 for experimental details.

**Backbone models**. As a model-agnostic framework, 3D-GeoCA is compatible with arbitrary backbone models, ranging from the basic multi-layer perceptron (MLP) to the GNO that follows the popular neural operator learning paradigm. In our work, the backbone model aims to solve PDEs in the problem domain $D$ and its boundary $\partial D$. It takes either the point cloud data $\mathbb{V}$ or graph data $\mathcal{G} = (\mathbb{V}, \mathbb{E})$ as input, where the vertex set $\mathbb{V} = \{\boldsymbol{x}_1^{\bar{D}}, \boldsymbol{x}_2^{\bar{D}}, \cdots, \boldsymbol{x}_N^{\bar{D}}\} \subset \bar{D} = D \cup \partial D$ contains all grids of interest. We also compute the SDF feature and normal vector to $\partial D$ for each vertex as a part of feature engineering. For the graph-based backbone model, the edge set $\mathbb{E}$ can be constructed according to the corresponding meshes or the radius graph with a maximum number of neighbors (Bonnet et al., 2022b). In this way, we can prevent the degree of each vertex from being too large and reduce the computation complexity.

**Geometry-guided conditional adaptor**. We propose a geometry-guided conditional adaptor, which enables the adaption of hidden features according to various geometries. At first, the adaptor conducts a feature fusion between hidden features in the backbone model and geometry features extracted by the point cloud encoder. Then, a feed-forward network processes the fused features to add non-linearity. Skip connections, normalization layers, and dropout layers are also added to our adaptor.

Denote $\boldsymbol{f}_{\text{hidden}}^{(l)}$ as the hidden features output by the $l$-th layer of the backbone model ($l = 1, 2, \cdots, L$), and $\boldsymbol{f}_{\text{geo}} = E_P(\mathbb{P})$ as the geometry features extracted by the point cloud encoder $E_P$. Formally, each adaptor can be formulated as follows:

$$\boldsymbol{f}_{\text{fused}}^{(l)} = \boldsymbol{f}_{\text{hidden}}^{(l)} + \text{norm}(\boldsymbol{f}_{\text{hidden}}^{(l)}) * \boldsymbol{W}_{\text{hidden}}^{(l)} + \text{norm}(\boldsymbol{f}_{\text{geo}}) * \boldsymbol{W}_{\text{geo}}^{(l)} \tag{2}$$

and

$$\boldsymbol{f}_{\text{adapted}}^{(l)} = \boldsymbol{f}_{\text{fused}}^{(l)} + \text{feedforward}^{(l)}(\text{norm}(\boldsymbol{f}_{\text{fused}}^{(l)})), \tag{3}$$

where $\boldsymbol{W}_{\text{hidden}}^{(l)}$ and $\boldsymbol{W}_{\text{geo}}^{(l)}$ are learnable parameters and $\text{norm}(\cdot)$ represents L-2 layer normalization. Equation 2 describes the process of feature fusion, by which we yield the fused feature $\boldsymbol{f}_{\text{fused}}^{(l)}$ of the $l$-th layer. In equation 3, $\boldsymbol{f}_{\text{fused}}^{(l)}$ are input into a feed-forward network to acquire adapted features $\boldsymbol{f}_{\text{adapted}}^{(l)}$ of the $l$-th layer.

For the structure of the feed-forward network, we select

$$\text{feedforward}^{(l)}(\boldsymbol{f}) = \text{GEGLU}^{(l)}(\boldsymbol{f}) * \boldsymbol{W}^{(l)} + \boldsymbol{b}^{(l)}, \tag{4}$$

where $\text{GEGLU}^{(l)}(\cdot)$ (Shazeer, 2020) is defined as

$$\text{GEGLU}^{(l)}(\boldsymbol{f}) = \text{GELU}(\boldsymbol{f} * \boldsymbol{W}_1^{(l)} + \boldsymbol{b}_1^{(l)}) \otimes (\boldsymbol{f} * \boldsymbol{W}_2^{(l)} + \boldsymbol{b}_2^{(l)}), \tag{5}$$

and $\boldsymbol{W}^{(l)}, \boldsymbol{b}^{(l)}, \boldsymbol{W}_1^{(l)}, \boldsymbol{b}_1^{(l)}, \boldsymbol{W}_1^{(l)}, \boldsymbol{b}_1^{(l)}$ are learnable parameters.

Finally, the adapted features $\boldsymbol{f}_{\text{adapted}}^{(l)}$ are fed into the $(l+1)$-th layer of the backbone model to get the hidden features $\boldsymbol{f}_{\text{hidden}}^{(l+1)}$ of the $(l+1)$-th layer.

Although our adaptor introduces additional structures to the backbone model, it requires only $O(h * (h + h_{\text{geo}}))$ parameters, where $h$ is the hidden size of the backbone model and $h_{\text{geo}}$ is the dimension of geometry features. Thus, our adaptor brings relatively low computational cost and inference latency once the backbone is a large model. As an example, an original 3D FNO layer with hidden size $h$ requires $O(h^2 M^3)$ parameters, where $M$ is the number of top Fourier modes being kept and usually be a large number, ranging from $O(10^1)$ to $O(10^2)$.

## 4 EXPERIMENTS

To empirically validate our findings, we conduct experiments on the public Shape-Net Car CFD Dataset generated by Umetani & Bickel (2018). Several previous works have explored this dataset (Umetani & Bickel, 2018; Li et al., 2023b), while Umetani & Bickel (2018) adopted the Gaussian process regression approach that falls outside the category of deep learning. Li et al. (2023b) also proposed the powerful GINO model for 3D PDEs with varying geometries, whereas their goal was to predict the pressure field at the boundary of the problem domain, namely $\partial D$. In contrast, we intend to simultaneously simulate plural physical properties (including both pressure and velocity) at all grids of interest in $\bar{D} = D \cup \partial D$. We trail multiple architectures of the point cloud geometry encoder and the backbone model, and all experiments can run on a single NVIDIA RTX A6000 GPU.

### 4.1 SHAPE-NET CAR CFD DATASET

The Shape-Net Car CFD Dataset was generated to study how fluid flows around various 3D objects (Umetani & Bickel, 2018). In that work, different object shapes of cars from the "car" category of ShapeNet Chang et al. (2015) were prepared, with their side mirrors, spoilers, and tries manually removed. The dataset contains 889 samples, each of which is a simulation result of a finite element solver. During the simulation, time-averaged fluid pressure on the surface and velocity field around the car was computed by solving the large-scale Reynolds Average Navier-Stokes equations with the $k$-$\epsilon$ turbulence model and SUPG stabilization. All the simulations ran with a fixed inlet velocity of 72 km/h and a Reynolds number of $5 \times 10^6$.

The dataset has already been randomly divided into nine folds. We take the first fold as our testing set, while the rest of the data consists of the training set. In each sample, the simulation result is discretized to 32k mesh grids, while the car surface counts merely 3.7k, implying that our backbone model and point cloud geometry encoder take 32k and 3.7k grids as input, respectively.

### 4.2 EXPERIMENTAL SETTINGS

**Backbone models**. Multiple architectures of the backbone model and the point cloud geometry encoder are employed to demonstrate the effectiveness of our framework. Since 3D-GeoCA is a

groundbreaking framework that correlates PDEs with the field of 3D understanding, we start with the simple MLP as our backbone model. Several classical GNNs, such as GraphSAGE (Hamilton et al., 2017) and Graph Attention Network (GAT) (Veličković et al., 2018) are also attempted in later. We also explore the application of 3D-GeoCA in the popular neural operator learning paradigm, where we consider GNO due to its ability to deal with irregular grid input directly.

**Point cloud geometry encoders.** As for the point cloud geometry encoder, we trial with two state-of-the-art point cloud architectures, Point-BERT (Yu et al., 2022) and PointNeXt (Qian et al., 2022). Point-BERT adopts transformer-based architecture, while PointNeXt is a lightweight backbone based on PointNet++ (Qi et al., 2017) with improved training and scaling strategies. Both point cloud models are pre-trained with the ULIP-2 framework on the Objaverse Triplets dataset (Xue et al., 2023b) to promote their capabilities to learn 3D geometry representations.

**Training schemes.** We normalize all inputs and outputs for data pre-processing. Since we target to predict the pressure and velocity by one forward propagation, we use the following weighted MSE loss to train models:

$$\text{Loss} = \frac{1}{N} \sum_{i=1}^{N} \left[ \frac{1}{n^{(i)}} \sum_{j=1}^{n^{(i)}} \|\boldsymbol{v}_{j,\text{pred}}^{(i)} - \boldsymbol{v}_{j,\text{gt}}^{(i)}\|_2^2 + \lambda \frac{1}{m^{(i)}} \sum_{\boldsymbol{x}_j^{(i)} \in \partial D} \|p_{j,\text{pred}}^{(i)} - p_{j,\text{gt}}^{(i)}\|_2^2 \right], \quad (6)$$

where $N$ denotes the number of training samples. $\boldsymbol{v}_j^{(i)}$ and $p_j^{(i)}$ represent velocity and pressure of the $i$-th sample at the $j$-th grid, respectively. $n^{(i)}$ denotes the number of input grids in the $i$-th sample, and $m^{(i)} = \sum_{j=1}^{n^{(i)}} \mathbf{1}_{\text{x}_j^{(i)} \in \partial \text{D}}$ is the number of boundary grids in the $i$-th sample. The hyper-parameter $\lambda$ balances the weight of the error between velocity and pressure, taking the default value of $0.5$.

We train our models with Adam optimizer and one-cycle learning rate scheduler (Smith & Topin, 2019). The batch size $B = 1$[1], and the maximum and minimum learning rates are $1 \times 10^{-3}$ and $1 \times 10^{-6}$, respectively. For the GNO backbone, the hidden size $h = 32$, and we train models for 200 epochs to save GPU memories and training times. Models of other backbones are trained for 400 epochs with the hidden size $h = 64$. For more implementation details, see appendix A.1.

**Evaluation metrics.** We introduce L-2 error and relative L-2 error to evaluate our models, which are defined as

$$\text{L-2 error} = \frac{1}{N} \sum_{i=1}^{N} \|\boldsymbol{u}_{\text{pred}}^{(i)} - \boldsymbol{u}_{\text{gt}}^{(i)}\|_2 \quad (7)$$

and

$$\text{relative L-2 error} = \frac{1}{N} \sum_{i=1}^{N} \frac{\|\boldsymbol{u}_{\text{pred}}^{(i)} - \boldsymbol{u}_{\text{gt}}^{(i)}\|_2}{\|\boldsymbol{u}_{\text{gt}}^{(i)}\|_2}, \quad (8)$$

where $\boldsymbol{u}$ represents the physical property of interest.

## 4.3 RESULTS

**The effectiveness of 3D-GeoCA.** Table 1 illustrates test L-2 errors of multiple backbone models with various point cloud geometry encoders. Since PointNeXt requires a training batch size greater than 1 to apply batch normalization, we keep its parameters frozen and do not fine-tune them. From table 1, we notice that 3D-GeoCA universally promotes all backbone models, reducing their L-2 errors by a large margin. For instance, with the trainable Point-BERT geometry encoder, MLP yields a marked descent in L-2 errors by $26\%$ and $37\%$ for pressure and velocity, respectively. The GNO [2], which follows the paradigm of operator learning, also benefits from our 3D-GeoCA framework, with its L-2 errors decreasing by $4\%$ for pressure and $18\%$ for velocity. By introducing a specialized geometry encoder, we take full advantage of the rich geometry information, and our adaptors enable backbone models to become more geometry-aware to generalize to unknown shapes.

---

[1] $B = 1$ implies we train the model on a batch of 32k graph nodes.

[2] We use a variant of GNO that utilizes hexahedral meshes generated by Umetani & Bickel (2018) to construct graphs. For original GNO, we also conduct experiments, see appendix A.2 for details.

| Geo. Encoder \ Backbone | MLP | | GraphSAGE | | GAT | | GNO | |
|---|---|---|---|---|---|---|---|---|
| None | 8.044 | 0.556 | 6.590 | 0.523 | 6.128 | 0.525 | 5.120 | 0.434 |
| PointNeXt (frozen) | 6.705 | 0.375 | 5.618 | 0.363 | 5.510 | 0.355 | 4.970 | 0.386 |
| | (-17%) | (-33%) | (-15%) | (-31%) | (-10%) | (-32%) | (-3%) | (-11%) |
| Point-BERT (frozen) | 6.456 | 0.368 | 5.630 | **0.349** | 5.629 | 0.346 | 4.991 | 0.365 |
| | (-20%) | (-34%) | (-15%) | (-33%) | (-8%) | (-34%) | (-3%) | (-16%) |
| Point-BERT (fine-tuned) | **5.916** | **0.352** | **5.569** | **0.349** | **5.438** | **0.339** | **4.906** | **0.356** |
| | (-26%) | (-37%) | (-15%) | (-33%) | (-11%) | (-35%) | (-4%) | (-18%) |

Table 1: Test L-2 errors of different backbone models with various geometry encoders. Errors of pressure is presented on the left side, while errors of velocity is presented on the right side. All errors are denormalized. Values in brackets represent the percentage of error reduction compared to the baseline with no geometry encoder.

Figure 2 visualizes a ground truth and prediction generated by the GNO backbone with the Point-BRET (fine-tuned) encoder. For more visualization examples, see appendix A.4. The inference of each sample costs 0.066 seconds, much faster than traditional numerical solvers. As a comparison, in their efforts to generate the Shape-Net Car dataset, Umetani & Bickel (2018) spent about 50 minutes per sample to run the simulations.

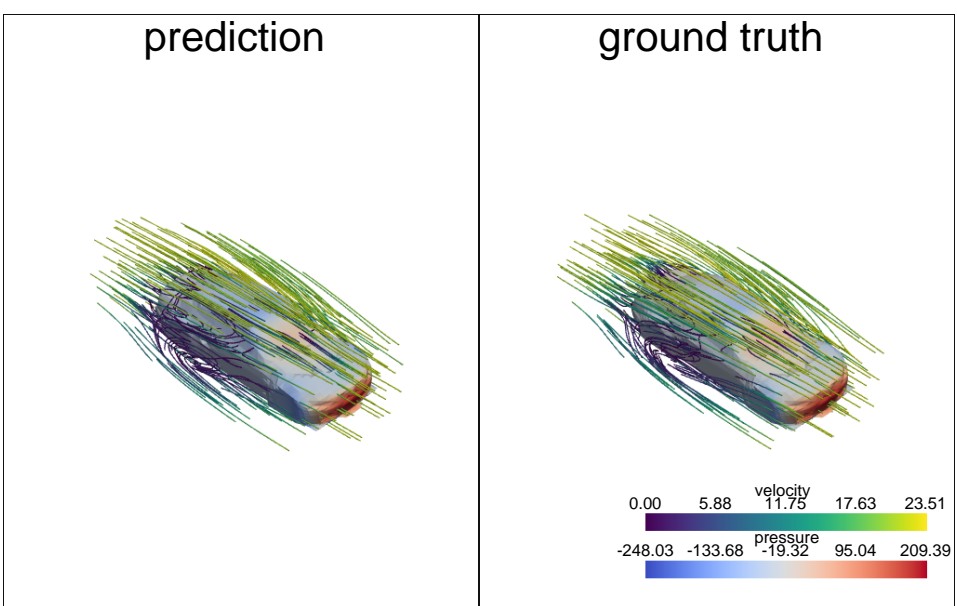

Figure 2: Visualization of a prediction and ground truth. The prediction is generated by the GNO backbone with Point-BRET (fine-tuned) encoder.

Moreover, 3D-GeoCA accelerates the convergence of the backbone model as well. Figure 3 exhibits the training loss of the GNO backbone with different geometry encoders for the beginning 20 epochs. The training loss of the GNO baseline decreases slowly, while the other three models equipped with 3D-GeoCA show higher convergence rates.

**Discussions on different geometry encoders**. Surprisingly, we observe that once pre-trained, even though the geometry encoder keeps frozen during training, it still extracts useful geometry information that can guide the adaption in the backbone model. As shown in table 1, the fixed PointNeXt and Point-BERT features pre-trained by ULIP-2 still perform well in the 3D-GeoCA framework and lead to competitive results compared to the fine-tuned features. This finding is of great significance,

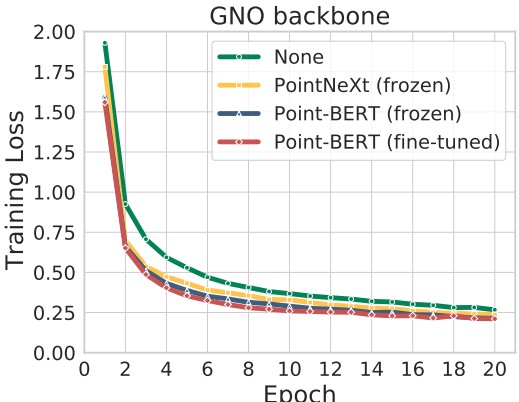

Figure 3: Training losses of GNO with different geometry encoders for the beginning 20 epochs.

implying that under our 3D-GeoCA framework, 3D understanding pre-training techniques may directly enhance the performance of PDEs surrogate models. Moreover, once the geometry encoder is frozen, we can pre-calculate geometry features and reduce the number of learnable parameters during training, further shortening the training process. Our GNO backbone with pre-trained Point-BERT features requires merely 0.6 million trainable parameters while reaching competitive low L-2 errors (relative L-2 errors) of $4.991$ ($7.91\%$) for pressure and $0.365$ ($3.23\%$) for velocity.

**The impact of various backbones**. As a novel framework that unprecedentedly correlates PDEs with the field of 3D understanding, we start with the simple MLP as our backbone model. Conceptually, according to the discussion in section 2, in steady-state equations with fixed parameters $a$ and boundary conditions $g$, the physical property $u$ at $(x, y, z)$ is merely decided by the geometry of problem domain $D$ and its coordinate $(x, y, z)$. However, as shown in table 1, the simple MLP gains relatively large L-2 errors, which might be because $u$ is a smooth function, and adjacent data points may share similar features. With the introduction of graph input structure, GNNs can learn better features from adjacent nodes via the message-passing mechanism. Consistent with our analysis, GraphSAGE and GAT perform better than MLP. Differing the above backbones, GNO aims to learn operators that map between function spaces via the kernel integration mechanism and shows an advantage in predicting pressure fields in our experiments. Moreover, according to the test loss, the overall performances of our models are positively correlated to the power of the applied backbone model.

**Comparisons with other works**. GNO with the trainable Point-BERT encoder achieves the lowest test loss in our experiments, with the L-2 error (relative L-2 error) of pressure and velocity of $4.906$ ($7.79\%$) and $0.356$ ($3.19\%$), respectively. As some reference values, the Gaussian Process Regression approach proposed by Umetani & Bickel (2018) reached a nine-fold mean L-2 error of $8.1$ for pressure and $0.48$ for velocity. The concurrent work GINO (Li et al., 2023b), submitted in the same month as our work, reported a relative L-2 error of pressure of $7.19\%$ for GINO (decoder) and $9.47\%$ for GINO (encoder-decoder). The relative L-2 error of GINO (decoder) is lower than that of our current experiments, though we should mention that their work has different training schemes and train-test split from ours. Another factor is that the GINO adopts a complex GNO-FNO architecture, while we have merely explored GNO as our backbone model. Moreover, the GINO can only simulate the pressure field at the surface of each car (3.7k grids). As the opposite, we train our models to simultaneously predict the velocity field around the car (32k grids) as well.

### 4.4 ABLATION STUDIES

**The selection of batch size**. During experiments, we find that the training batch size is the most critical hyper-parameter that impacts the quality of training. Table 2 shows the test L-2 errors under different batch sizes [3], from which we observe that using a larger batch size during training brings a universal negative influence on our models, especially for simple architectures, such as models

---

[3]We also fine-tune PointNeXt under $B = 2$, see appendix A.3

with MLP backbone and models with no geometry encoder. When the model structure becomes complex, there is relatively less increase in test L-2 error. Although it is more obvious to validate the effectiveness of our 3D-GeoCA framework when the batch size $B = 2$, we set $B = 1$ in our experiments to ensure that every model obtains the best results.

| Geo. Encoder / Backbone | MLP | | GraphSAGE | | GAT | | GNO | |
|---|---|---|---|---|---|---|---|---|
| None (bs=1) | **8.044** | **0.556** | **6.590** | **0.523** | **6.128** | **0.525** | **5.120** | **0.434** |
| None (bs=2) | 9.976 | 0.688 | 7.470 | 0.611 | 6.957 | 0.622 | 5.872 | 0.517 |
| PointNeXt (frozen, bs=1) | **6.705** | **0.375** | **5.618** | **0.363** | **5.510** | **0.355** | **4.970** | **0.386** |
| PointNeXt (frozen, bs=2) | 8.758 | 0.546 | 6.293 | 0.467 | 6.273 | 0.471 | 5.581 | 0.479 |
| Point-BERT (frozen, bs=1) | **6.456** | **0.368** | **5.630** | **0.349** | **5.629** | **0.346** | **4.991** | **0.365** |
| Point-BERT (frozen, bs=2) | 7.796 | 0.437 | 5.909 | 0.411 | 5.922 | 0.399 | 5.329 | 0.411 |
| Point-BERT (fine-tuned, bs=1) | **5.916** | **0.352** | **5.569** | **0.349** | **5.438** | **0.339** | **4.906** | **0.356** |
| Point-BERT (fine-tuned, bs=2) | 6.689 | 0.423 | 5.571 | 0.360 | 5.454 | 0.351 | 4.957 | 0.374 |

Table 2: Test L-2 errors under different batch sizes. Errors of pressure is presented on the left side, while errors of velocity is presented on the right side. All errors are denormalized.

**The robustness of geometry encoders.** We further demonstrate the robustness of our fine-tuned geometry encoder by randomly dropping its input data points in the inference time. Figure 4 depicts how L-2 errors varied with the rate of dropping increases. Surprisingly, even if we drop each data point with a high probability of $60\%$, our encoder still extracts essential geometry features that can guide the adaption of hidden features in the backbone model, demonstrating its strong robustness.

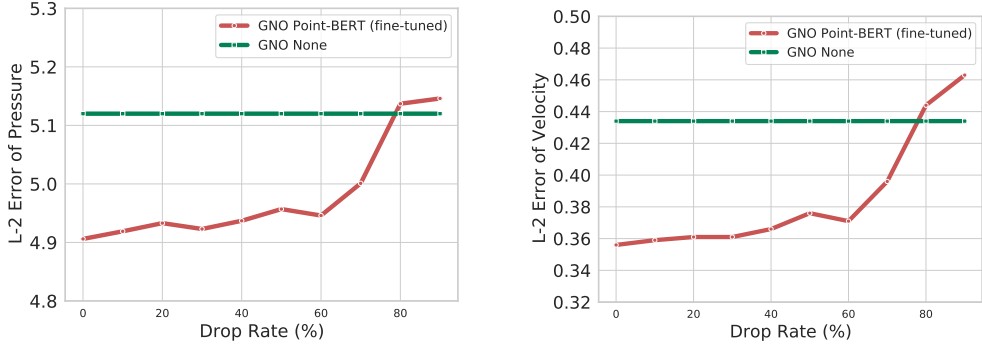

Figure 4: Inference with input of geometry encoder randomly dropped.

## 5 CONCLUSION

Learning the solution of 3D PDEs with varying geometries is challenging due to the complexity of 3D shapes and insufficient training samples. By introducing a specialized point cloud geometry encoder, our proposed 3D-GeoCA framework learns essential and robust geometry features that can guide the adaption of hidden features in the backbone model. 3D understanding pre-training further enhances our framework. Several backbones reduce L-2 errors by a large margin equipped with 3D-GeoCA. So far, our way of conditional adaption remains simple and may not be optimal. For future work, we are interested in exploring other effective and efficient structures for our adaptor. Moreover, we expect our framework to be compatible with the backbones of broader fields, such as FNO and Physics-Informed Neural Network (PINN) (Raissi et al., 2019).

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

## A APPENDIX

### A.1 IMPLEMENTATION DETAILS

All backbones contain $L = 3$ hidden layers with sizes of $h$. In our implementations, each backbone is preceded by an MLP with $7 - 32 - h$ neurons to ascend the dimension of hidden features and followed by an MLP with $h - 32 - 4$ neurons to perform regression. We use GELU as activation function.

As for the feed-forward network, we select $\boldsymbol{W}_1^{(l)}, \boldsymbol{W}_2^{(l)} \in \mathbb{R}^{h \times 4h}$, $\boldsymbol{b}_1^{(l)}, \boldsymbol{b}_2^{(l)} \in \mathbb{R}^{4h}$ and $\boldsymbol{W}^{(l)} \in \mathbb{R}^{4h \times h}$, $\boldsymbol{b}^{(l)} \in \mathbb{R}^h$. In addition, in our geometry-guided conditional adaptor, $\boldsymbol{W}_{\text{hidden}}^{(l)} \in \mathbb{R}^{h \times h}$ and $\boldsymbol{W}_{\text{geo}}^{(l)} \in \mathbb{R}^{h_{\text{geo}} \times h}$. Overall, each adaptor requires $13h^2 + hh_{\text{geo}} + 9h = O(h(h + h_{\text{geo}}))$ parameters.

For MLP, GraphSAGE and GAT backbones, $h = 64$. For the GNO backbone, $h = 32$. In accordance with (Xue et al., 2023b), the size of geometry feature $h_{\text{geo}} = 512$.

### A.2 GNO AND ITS VARIANT

The original GNO backbone implements the kernel integration by considering neighbor nodes lie in a ball of radius $r$. During the message passing, a shared kernel network encodes the attribute of each edge to a $h^2$-dimension feature to determine the corresponding weight of feature aggregation.

However, when $r$ is large, numerous edges are needed to process, and the whole training procedure becomes memory-consuming. Instead, we use a GNO variant that constructs graphs according to the hexahedral mesh generated by Umetani & Bickel (2018) (which is the same way as we construct graphs for GNNs in this work) and yields low L-2 errors. For the original GNO, we also conducted experiments. We choose radius $r = 0.2$ and a maximum number of neighborhoods to be 32 to construct radius graphs. The average number of edges of these graphs roughly matches that of our mesh-based graphs. Figure 3 shows the experimental results, where we still observe the effectiveness of 3D-GeoCA when radius graphs are adopted, while the GNO backbone performs worse than our GNO variant.

| Geo. Encoder \ Backbone | GNO (radius) | | GNO (mesh) | |
|---|---|---|---|---|
| None | 5.546 | 0.465 | **5.120** | **0.434** |
| PointNeXt (frozen) | 5.189 | 0.405 | **4.970** | **0.386** |
| Point-BERT (frozen) | 5.298 | 0.399 | **4.991** | **0.365** |
| Point-BERT (fine-tuned) | 5.212 | 0.373 | **4.906** | **0.356** |

Table 3: Test L-2 errors of the GNO backbone and its variant. Errors of pressure is presented on the left side, while errors of velocity is presented on the right side. All errors are denormalized.

### A.3 FINE-TUNING POINTNEXT

Table 4 shows test L-2 errors of the GNO backbone under batch size of 2. Since PointNeXt requires a training batch size greater than 1 to apply batch normalization, we only fine-tune it under batch size $B = 2$.

| Geo. Encoder \ Backbone | GNO | |
|---|---|---|
| None (bs=2) | 5.872 | 0.517 |
| PointNeXt (frozen, bs=2) | 5.581 | 0.479 |
| PointNeXt (fine-tuned, bs=2) | 5.530 | 0.472 |
| Point-BERT (frozen, bs=2) | 5.329 | 0.411 |
| Point-BERT (fine-tuned, bs=2) | 4.957 | 0.374 |

Table 4: Test L-2 errors of the GNO backbone under batch size of 2.

### A.4 VISUALIZATION

In this section, we provide more visualization examples. All predictions are generated by the GNO model with the Point-BERT (fine-tuned) encoder.

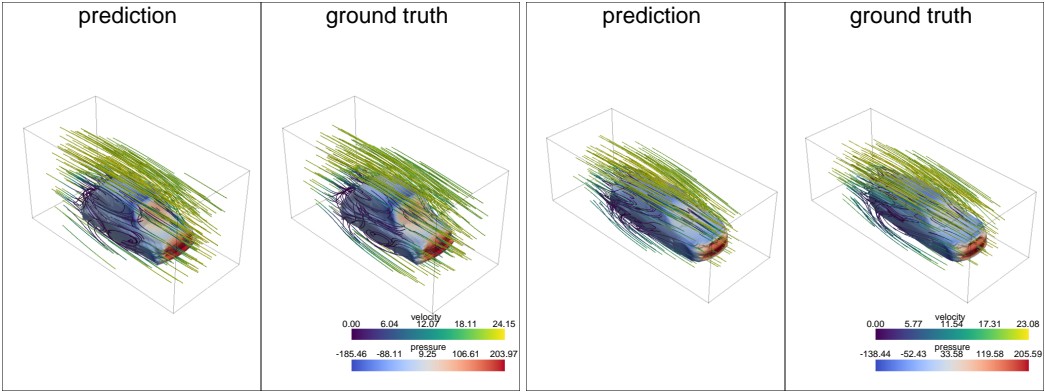

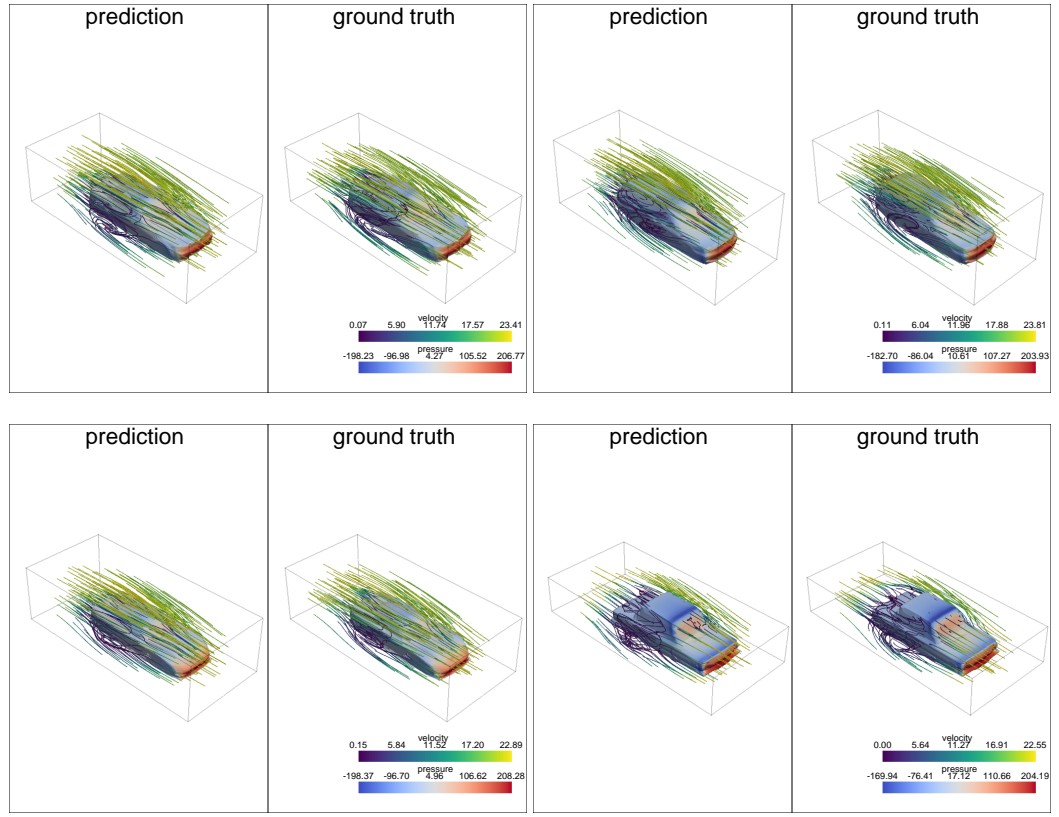

## A.5 Detailed Information about Figure 2

Figure 5 provides detailed information about Figure 2. Since it is hard to visualize the difference of velocity fields, only relative L2 error and difference of pressures are visualized.

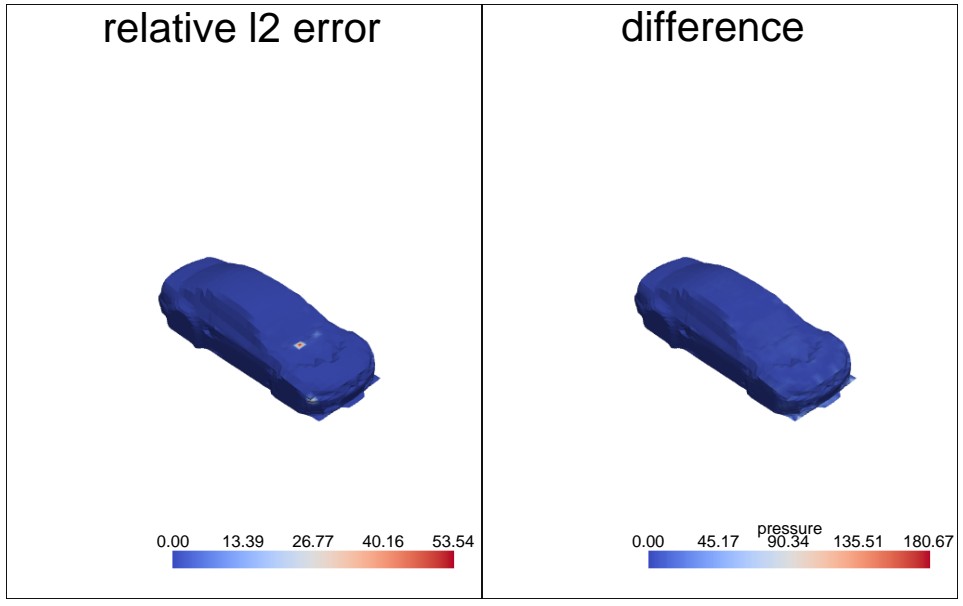

Figure 5: Detailed information about Figure 2.

