# OpenReview forum: "Geometry-Guided Conditional Adaption for Surrogate Models of Large-Scale 3D PDEs on Arbitrary Geometries"
_ICLR.cc/2024/Conference — ICLR 2024 Conference Withdrawn Submission_

### Official Review · Reviewer_SunQ · 2023-10-24

**Soundness:** 2 fair
**Presentation:** 2 fair
**Contribution:** 3 good
**Rating:** 5
**Confidence:** 2

**Summary:**

The paper proposes 3D-GeoCA, a method to solve PDE on arbitrary geometry. It makes use of a geometry encoder that extracts features from the point cloud, a backbone network that processes the geometry as a point cloud or graph; and an adaptor that merges these two sets of features.

**Strengths:**

The method addresses an important problem that has a potential impact on different fields. Even if it is designed for physical simulation, I understand that it can provide a better understanding of general geometries and so provide a contribution to Computer Graphics and Vision communities. The paper seems to provide a good discussion of recent literature, also considering contemporary works.

**Weaknesses:**

I am not an expert in this specific field, and hence I may overlook some important aspects. Given this, I have the following concerns:

1) CLARITY: I went through the method section a couple of times, but still, I am not able to fully understand the proposed approach and the specific insight of the method. In particular, I find the paragraph about adaptors a bit cluttered with technical and implementation details, which hide the flow of the proposed approach. The output of different steps is not clear, and figure 1 lacks a proper caption. As a nonexpert, I do not feel I can make use of or re-implement the method and what are the main insights, and I think this is a significant limit in the potential impact of the work.

2) COMPARISON: About the experiments, Figure 2 and the qualitative results in the appendix all provide a comparison between the result and the method, but without a baseline, it is difficult to assess the quality of the obtained results. I suggest the authors to include more visualization to help unexperienced readers to get into the topic and better understand the intuition

**Questions:**

1) What is the computational timing of the proposed approach? How much does the complexity of the input geometry affect this?
2) If I understand correctly, the provided ablation shows the results about dropping input data points at inference time. The method performance does not degrade much until it drops 60% of the points. Does it mean the geometry encoder suffers from partiality, e.g., missing part of the model? Does it rely on the assumption that the shape at inference time has more or less the same geometry of the training distribution, while it can be more sparse?
3) The method uses pointnet-based feature extractors. However, why not consider more recent options that could provide more structured inductive bias (e.g., DeltaConv: Anisotropic Operators for Geometric Deep Learning on Point Clouds, Wiersma et al., 2022)?

---

> ### Author Response · Authors · 2023-11-21
>
> Thank you for your review and constructive comments. We believe all your comments could greatly help us improve the manuscript. Here we would like to clarify some major concerns in your comments.
> 1. Thanks for your recommendation. We have revised our manuscript. The main insight of 3D-GeoCA is that we emphasize the geometries of various PDE problem domains and introduce a specialized geometry encoder to learn them. Furthermore, we propose an adaptor to integrate existing surrogate backbone models with the learned geometry features. The adaptor is designed in a feature fusion way to allow the interaction of geometry features and hidden features in the backbone surrogate model. All the code and data are available.
> 2. In Table 1 and Table 2, the rows annotated with the “None” geometry encoder represent the original existing surrogate models (i.e., MLP, GraphSAGE, GAT, and GNO). These models are our baseline. By introducing different types of geometry encoders (i.e., PointNext and Point-BERT, either frozen or fine-tuned), our 3D-GeoCA universally promotes all backbone models, reducing their L-2 errors by a large margin.
>
> Response to Q1. We have mentioned that the GNO backbone with the Point-BRET (fine-tuned) encoder infers each sample within 0.066 seconds on page 7, which is fast enough. As for training time, the GNO backbone with the PointNeXt (frozen) encoder requires merely 17% extra training time compared to the original GNO, while the GNO backbone with the Point-BERT (fine-tuned) encoder requires an additional 50% time compared to the original GNO.
>
> Response to Q2. Yes. We need to assume the shape at inference time has more or less the same geometry of the training distribution.
>
> Response to Q3. PointNeXt and Point-BERT were also proposed in 2022. We chose these two feature extractors because the adopted pre-training method, ULIP-2, was experimented on them.

---

> > ### Comment · Reviewer_SunQ · 2023-11-22
> > **post-rebuttal**
> >
> > I thank the Authors for their reply. The answer addressed some of my concerns, while the major weakness of the clarity (which has been indicated also by other reviewers) remains quite critical (I still have trouble fully appreciating the contribution). I also see the point of Reviewer vtzc about the experimental setting. At the moment, I am quite hesitant to raise my score. However, both I and reviewer vtzc mentioned to be not experts in the field. I am looking forward to knowing other reviewers' opinions and engaging in a discussion with them.

---

### Official Review · Reviewer_vtzc · 2023-10-31

**Soundness:** 2 fair
**Presentation:** 2 fair
**Contribution:** 3 good
**Rating:** 3
**Confidence:** 2

**Summary:**

This paper proposes the 3D-GeoCA method, a framework that aims to leverage learned geometry representations from existing point cloud models to solve PDEs. The approach is generic and works with different backbone neural PDE models. The authors evaluate the method on the shape-net car CFD dataset and report improvements of up to 10-30% across different backbone models. Ablations on different point cloud models and model batch sizes are included as well.

**Strengths:**

This paper has two main strengths, originality and promising results.

Firstly, to me the proposed framework is quite original. Combining 3D shape representations with neural PDE solvers in such a generic way is an interesting direction.

Secondly, the shown results indicate that there seems to be a consistent 10-30% decrease in prediction error when employing the proposed framework across different backbone models.

Furthermore, source code is provided with this submission, which helps to improve reproducibility in the future. However, I did not attempt to run the code or investigate it in detail.

**Weaknesses:**

I am not an expert in the domain that this work targets, and I am unfamiliar with a range of the cited related work. As a result, I was struggling to fully understand the details of the proposed approach. Nevertheless, to me this paper appears unfinished in various aspects.

### Presentation

**P1:**
There are several references to related work in the introduction, but in my opinion this paper would benefit from a dedicated related work section. Furthermore, the paper does not address a substantially amount of related work targeting PDE solutions via graph networks. For instance, some key papers from this domain include:
- *“Learning to simulate complex physics with graph networks”* by Sanchez-Gonzalez et al., ICML 2020
- *“Message passing neural PDE solvers”* by Brandstetter et al., ICLR 2022
- *“Combining differentiable PDE solvers and graph neural networks for fluid flow prediction”* by de Avila Belbute-Peres et al., ICML 2020
- *“Learning mesh-based simulation with graph networks”* by Pfaff et al., ICLR 2021

**P2:**
Several statements in this work are vague or unclear, and require additional explanation or citations. The following list is not exhaustive, but some example include:
- “can extract the essential and robust representations of any kind of geometric shapes, which is regarded as a conditioning key to guiding the adaption of hidden features in the surrogate model.” (unclear, abstract)
- “Actually, various approaches have been proposed, including finite difference and finite element methods, whereas these methods usually have high computational costs, which are unendurable in many real-time settings.” (lacks citation, first paragraph in section 1)
- “While the work above has achieved remarkable progress in solving simple 2D equations” (even in 2D the investigated PDEs can create complex behavior, paragraph 4 in section 1)
- Inefficient Representation for 3D Inputs section: Choosing non-Cartesian meshes in the context of graphnet-based PDE simulators (as mentioned above) does exactly address this problem, if I understood the argumentation here correctly?
- “Current work usually under-emphasizes grids in P in boundary of D and coarsely feeds all grids with simple hand-crafted features (such as SDF) into their model” (lacks citation, paragraph 3 in section 3)

**P3:**
The paper lacks overall polishing in terms of language, notation, formatting and figure design.

### Evaluations

**E1:**
The proposed method is quite generic, and as such it should also be evaluated in a generic way, however only a limited scope of experiments on a single data set are shown. Ideally, at least one more complex case from a fluid flow perspective could be considered, for example unsteady flow problems. Furthermore, generalization ability is highly desirable for ML-based PDE solvers, and thus evaluations on test sets outside of the direct training domain (for example different Reynolds numbers) should be considered, not only a random train-test split. Similarly, only a single evaluation metric via a simple L2 distance is used. Especially the domain of fluid flows offers a range of tools for more in-depth evaluations, for example spectral analyses could be considered (see e.g. *“Turbulent flows”*, Pope, Cambridge University Press).

**E2:**
The proposed architecture adds a substantial amount of computational and expressive complexity to the backbone networks. This directly raises the question if additional model capacity chosen in a suitable manner (e.g. number of parameters, GPU memory, overall training time, etc.) would not increase the backbone performance just as much. Furthermore, a comparison with a model that uses the same architecture but does not pretrain the geometry encoder would be necessary, to draw conclusions regarding the usefulness of pre-training.

**E3:**
To me it seems problematic that the pressure and velocity errors in Tab. 1 and 2 differ so much. Why would the models work so much better in predicting velocity over pressure? I assume this might be an issue with the data ranges, when looking at the colorbars in Fig. 2. If the pressure has a substantially higher magnitude, it should be normalized separately such that the network has a chance to determine both with similar accuracy.

**E4:**
Drawing conclusions on the convergence rate of different models in Fig. 3, does not seem ideal. Considering that the models are trained for 200 epochs, only looking at the first 20 epochs is a limited evaluation. This also ties into the question of larger model capacities mentioned in **E2** above.

### Summary
Overall, the results of this paper are pointing in an interesting direction and the approach is quite original. Nevertheless, the weaknesses of this work are more dominant than its strengths. The presentation appears unfinished, the scope of the experiments is limited, and there are some problems with the shown evaluations. This leads to my overall recommendation of reject for the current state of this paper.

**Questions:**

**Q1:**
I am curios about the choice of $\lambda=0.5$ in equation 6. This seems to arbitrarily bias the network to either focus on pressure or velocity instead of treating them in the same way (depending on the normalization of the input fields). My guess is that this also impacts the pressure vs. velocity error issue discussed in **E3** above.

**Q2:**
Why is the evaluation in Fig. 4 meaningful? Dropping data points from the encoder is an artificial operation that should not happen in practice. And I would interpret the right half of each plot as a point against the proposed architecture: even though the model is trained to heavily rely on the geometry encoder, it only performs marginally worse when having barely any information from the encoder at all during inference.

---

> ### Author Response · Authors · 2023-11-21
>
> Thank you for your review and constructive comments. We believe all your comments could greatly help us improve the manuscript. Here we would like to clarify some major concerns in your comments.
>
> P1. There are various types of PDEs with different properties. In this paper, we mainly focus on a family of PDEs: 1. with varying domains; 2. in 3D space. However, [1,2] are not involved in PDEs with varying computational domains and hence not highly relevant to our work. [3,4] have studied two airfoil datasets, but they are both 2D. We have revised our manuscript and include papers [3,4].
>
> P2. (clarification of “Inefficient Representation for 3D Inputs”) It is not the introduction of non-Cartesian meshes that addresses this problem, but the original introduction of the point cloud geometry encoder that encodes the boundary of the problem domain. Most existing approaches treat each position in the field equally and coarsely feed all grids into the model. However, this may increase the difficulty of learning geometry features of the problem domain, since the boundary points are actually much more informative than other points for PDE solving. We have clarified this point in our revised manuscript.
>
> P2-P3. We have revised our manuscript. Thank you very much for all the corrections about our presentations.
>
> E1. (dataset) Most of the public datasets that describe PDEs with varying domains are 2D, and generating a large-scale 3D PDEs dataset is time-consuming and computationally expensive. Therefore, it is not easy to find another suitable dataset for our research up to now. Currently, we mainly focus on the steady-state equations, and we will devote ourselves to generating and exploring an unsteady-state flow dataset in the future.
>
> E1. (metrics) Admittedly, the L2 and relative L2 errors used in our paper have their limitations as aggregate metrics. However, there is no direct way to apply the Fourier transform on the irregular mesh to conduct spectral analysis. Therefore, we just follow previous work [3,4,5,6,7] and use L2 and relative L2 errors.
>
> E2. (architecture complexity) Our 3D-GeoCA is designed as simple as possible. It does not introduce many learnable parameters and computational complexity: 1. The geometry encoder only takes boundary points as input, which is a tiny part of the whole input; 2. The geometry encoder can be frozen and hence geometry features can be pre-calculated during training; 3. The conditional adaptor (whose hidden size is designed to equal the hidden size of the backbone model) uses the simplest additive feature fusion. For instance, introducing the PointNeXt (frozen) encoder to the GNO backbone merely increases 17% training time (even if we did not pre-calculate geometry features), implying that we do not intend to blindly strengthen the additional model to yield good results.
>
> E2. (pre-training) Thank you for your recommendation. Our 3D-GeoCA is effective even with the FROZEN pre-trained geometry encoder, which is even stronger evidence that proves the usefulness of pre-training in our opinion. If the pre-trained geometry feature is useless and irrelevant to this task, no promotions will be observed in this experimental setting.
>
> E3. All the results in our table are denormalized. We have clarified this in our revised manuscript. However, as mentioned in the training schemes (on page 6), we normalize all the inputs and outputs when training our models.
>
> E4. Thanks for your opinion.
>
> Q1. It is an empirical value that largely reduces the L2 error of pressure while the L2 error of velocity does not increase much compared to \lambda=1.
>
> Q2. We care about the scalability of our method-- different from CV or NLP, one can simulate PDEs arbitrarily fine, and the input size can be extremely large. However, the current graph/point cloud model usually takes O(10^4) or fewer points as input, implying that a process of subsampling is necessary. The evaluation in Fig.4 ensures that the learned geometry feature is robust during this operation.
>
> [1]“Learning to simulate complex physics with graph networks” by Sanchez-Gonzalez et al., ICML 2020
>
> [2]“Message passing neural PDE solvers” by Brandstetter et al., ICLR 2022
>
> [3]“Combining differentiable PDE solvers and graph neural networks for fluid flow prediction” by de Avila Belbute-Peres et al., ICML 2020
>
> [4]“Learning mesh-based simulation with graph networks” by Pfaff et al., ICLR 2021
>
> [5]“Multipole graph neural operator for parametric partial differential equations” by Zongyi Li et al., Advances in Neural Information Processing Systems 2020
>
> [6]“Fourier neural operator for parametric partial differential equations” by Zongyi Li et al., ICLR 2020
>
> [7]“Factorized fourier neural operators” by Alasdair Tran et al., ICLR 2022.

---

> > ### Comment · Reviewer_vtzc · 2023-11-22
> > **Thanks for your response**
> >
> > I would like to thank the authors for their response, and I had a look at the revised version of the paper. The comments and changes address some of my concerns, especially with regard to **P1** and **P2**. Nevertheless, I still believe this paper requires more extensive and thorough experiments, as mentioned in **E1** and **E2**. For instance, even data sets or metrics on regular grids would be an option as a first step, by simply resampling them to irregular meshes as a toy example. Furthermore, a generally improved presentation is necessary to improve clarity, as also mentioned by reviewers SunQ and E9jv.
> >
> > These weaknesses are fundamental enough for me to not recommend a higher overall score for the current state of this work.

---

### Official Review · Reviewer_E9jv · 2023-10-31

**Soundness:** 3 good
**Presentation:** 2 fair
**Contribution:** 2 fair
**Rating:** 5
**Confidence:** 3

**Summary:**

Paper introduces a framework for predicting outputs of 3D PDEs that operates directly on geometry (point clouds).
The key idea is to combine a generic pre-trained geometry encoder, combine with state-of-the-art prediction backbone and train resulting model to predict the results of ground truth simulations. The method is robust to the choice of the underlying backbone (whether it's MLP or a graph CNN), and demonstrates comparable performance to some an existing baseline.

**Strengths:**

- The work is well-motivated: classical CFD and similar kinds of simulations are expensive, and thus developing efficient and generalizable surrogate models has huge potential.
- The overall approach of using a pre-trained geometry encoder features to improve generalization makes a lot of sense.
- Proposed framework is agnostic to the choice of the backbone and seems to be producing accurate predictions (although there are questions to evaluation, see below).

**Weaknesses:**

- (Novelty) Using a geometry encoder for predicting outputs of physical simulations, in particular with mesh-CNNs have been used before [Baque'18].
- (Clarity) It is a bit hard to understand what are the actual contributions of this work. The terminology provided in the into is quite confusing: is the main novelty that the method relies on pre-trained geometry decoder? Is it the fact that it directly operates on geometry rather than ad-hoc parameterizations? Is it the specific adapter architecture that is fundamental? This needs clarification.
- (Evaluation) Since the speed is one of the core reasons for using surrogate models over classical simulation, it is surprising that no formal evaluation of speed have not been conducted.
- (Evaluation) There are classical methods (GPs aka kriging) which are (still) standard in the engineering fields. It would probably make sense to compare against those?

Typos / misc:
- I am not a native speaker, but "adaption" seems like a misspelling of "adapter" / "adaptation"?

**Questions:**

- Would be great if you could clarify the contributions and specifically technical novelties, currently all four points in the intro sound bit like the same thing rephrased multiple times.
- The presentation of results in Section 4.3 is a bit weird: do not see why not present results in the same table? Why not compare to existing tools e.g. GPs?
- One of the key benefits of using neural nets as surrogate models are the fact that they are differentiable [Baque'18], and are suitable e.g. for shape optimization. Did you consider using the resulting model for that purpose? Do you foresee any issues for any of the chosen backbones?

---

> ### Author Response · Authors · 2023-11-21
>
> Thank you for your encouraging review and constructive comments. We believe all your comments could greatly help us improve the manuscript. Here we would like to clarify some major concerns in your comments.
> 1. (The novelty and main contribution of 3D-GeoCA) Our goal is different from that of Baque et al.[1]. Aside from predicting physical properties on the surface, we also aim to simulate the other property (velocity field) around the shape. Most existing approaches (with the same goal as ours), though some used point-cloud-based models, treated these input positions equally. However, the geometry (represented by boundary points) is actually much more informative than other points for PDE solving. Therefore, we introduce a specialized geometry encoder and propose a conditional adaptor to take full advantage of the rich geometry information. We have re-organized our contributions in our revised manuscript.
> 2. (The speed of surrogate model) We have mentioned that the GNO backbone with the Point-BRET (fine-tuned) encoder infers each sample within 0.066 seconds on page 7, which is much faster than traditional numerical solvers. As a comparison, in their efforts to generate the Shape-Net Car dataset, Umetani and Bickel [2] spent about 50 minutes per sample to run the simulations.
> 3. (The evaluation of GPs) As we mentioned on page 8, a previous work [2] has studied the same dataset using Gaussian Process Regression, reaching a nine-fold mean L-2 error of 8.1 for pressure and 0.48 for velocity. Since our results are not nine-fold mean errors, we do not list their results in our table.
> 4. (Typos) “Adaption” is another version of “adaptation” that is not commonly used.
>
> Response to Q1: Thank you for your recommendation. We have revised our manuscript.
>
> Response to Q2: Some results from other work are not derived under the same experimental setting as ours. Listing them in the same table may cause misleading.
>
> Response to Q3: Currently, we aim to learn the solution of 3D PDEs with varying geometries. Shape optimization is an important inverse problem and has extensive applications, yet we have not designed an algorithm for it.
>
> [1]Baque, Pierre, et al. "Geodesic convolutional shape optimization." International Conference on Machine Learning. PMLR, 2018.
> [2]Nobuyuki Umetani and Bernd Bickel. Learning three-dimensional flow for interactive aerodynamic design. ACM Transactions on Graphics (TOG), 37(4):1–10, 2018.

---

> > ### Comment · Reviewer_E9jv · 2023-12-04
> >
> > I would like to thank authors for the reply. This answers some of my concerns.
> > For 2., I don't think the comparison should be wrt to the numerical solvers, but rather to the baselines. There are already ML-based surrogate model approaches that operate on meshes/point clouds.
> > It looks like there is a large concern for the clarity / presentation quality of the paper - given that it is rather uniform across reviewers - I am inclined to lower my score.

---

### Official Review · Reviewer_Wkj4 · 2023-11-01

**Soundness:** 3 good
**Presentation:** 3 good
**Contribution:** 2 fair
**Rating:** 5
**Confidence:** 3

**Summary:**

This paper proposes to use 3D point encoder architectures pre-trained on large-scale 3D shape datasets to extract global point cloud features and to use these features to condition the neural network models designed for solving PDEs in 3D space. The conditioning mechanism adds the extracted global point cloud features put through an MLP to per-point features computed by a separate backbone network for every input grid or surface point.

In the experiments, the authors vary several architectures for pretrained global feature extractors and backbone networks and evaluate the approach by calculating the mean absolute/relative L2 distance between the ground truth and predicted physical property values at input locations. Results suggest that global conditioning with pre-trained and possibly fine-tuned features improve the quality of the inferred PDE solutions.

**Strengths:**

I do not possess extensive knowledge of the literature, but the authors claim that this is the first attempt at using geometric feature extractors applied to surface samples to condition the PDE solution prediction networks.

**Weaknesses:**

1. The choice of conditioning mechanism seems arbitrary. General purpose feature conditioning is an established field in deep learning, the closest to the proposed mechanism is FiLM: Visual Reasoning with a General Conditioning Layer. Since then various adaptor schemes have been applied to any imaginable learning task across numerous domains, e.g. conditioning on global features and time-positional features in diffusion models Scalable Diffusion Models with Transformers. I think the paper can benefit greatly from considering at least several options for conditioning mechanisms and comparing them.

2. No intuition is provided to explain the degradation in quality when the model is trained with batch size > 1. I suspect there may be a technical error in the batched implementation of the approach causing this, rather than a consistent result, which will translate to other methods.

3. I do not know if this is the norm for this task, but using mean L2 distance as the only evaluation metric is limiting. As it is an aggregate metric, it is impossible to assess the consistency/smoothness of the produced solution, and the presence of the outliers. At least adding the variance/std can show something. Also, maybe a heatmap of L2 distance instead of the heatmap of the solution can show where the model is the least accurate.

**Questions:**

I do not expect the authors to solve all this, so I put some additional comments in the questions section:

1. Why no positional encoding for input locations?

2. Why do you use global conditioning for all the input points? You can use local per-point features at least for points on the surface. And even for the grid points it is possible to obtain some features by discretising the point features onto the grid and applying convolutions to spread the features across the whole grid, e.g. point-cloud to 3D grid network from Neural Dual Contouring.

3. I don’t know if other metrics are established for the task, but L2 as a final metric is a bit misleading, in my opinion. The end goal of the considered simulations is to make a decision about the shape of the car and the materials, that will be able to deal with the pressure. Do you need as precise simulations as possible to make this decision? How much does this L2 error translate into incorrect choices for the shape and materials? I believe these things are quantifiable and I will be very interested to see metrics that can evaluate this, rather than simple final optimized loss value.

---

> ### Author Response · Authors · 2023-11-21
>
> Thank you for your review and constructive comments. We believe all your comments could greatly help us improve the manuscript. Here we would like to clarify three major concerns in your comments.
> 1. Thanks for your precious recommendation on the choice of conditioning mechanisms. Currently, we use the additive feature fusion approach for simplicity, keeping in mind not to introduce too many learnable parameters and extra computational costs compared to the original backbone model. As we mentioned in the conclusion, although our way of conditional adaption is effective, it remains simple and may not be optimal. We will devote ourselves to exploring other effective and efficient structures for our adaptor for future work.
> 2. The batch size in our paper equals the number of 3D shapes in a batch. However, from the perspective of regression tasks, the batch size of 1 also implies we train the model on a batch of 32k graph nodes (as we stated in the footnote of page 6), i.e., there are 32k data points for supervision in a batch, which is large enough. On the other hand, if we further increase the batch size, the number of iterations reduces, which may cause insufficient updates of parameters and harm the training effectiveness. We have checked our implementation and confirmed our results.
> 3. Thank you for pointing out that. We have visualized the differences between the prediction and ground truth in the appendix. Admittedly, the L2 and relative L2 errors used in our paper have their limitations as aggregate metrics, but currently, this field lacks an appropriate metric to evaluate the consistency/smoothness of the generated solution. Therefore, we follow previous work [1,2,3,4] and use L2 and relative L2 errors. Several metrics can help decision-making more directly. Unfortunately, they are task-specific but not general (For instance, lifting coefficients derived from velocity and pressure fields help design an airfoil).
>
> Response to Q1: We take the coordinates (x, y, z) as a part of input features, hence probably there is no need to apply positional encoding.
>
> Response to Q2: Using local per-point features is a good idea and allows us to attempt more powerful conditioning mechanisms, such as cross-attention. However, we have little concern that this method may be relatively computationally expensive and cannot scale up -- different from CV or NLP, one can simulate PDEs arbitrarily fine, and the input size can be extremely large.
>
> Response to Q3: See 3.
>
> [1] Anima Anandkumar, Kamyar Azizzadenesheli, Kaushik Bhattacharya, Nikola Kovachki, Zongyi Li, Burigede Liu, and Andrew Stuart. Neural operator: Graph kernel network for partial differential equations. In ICLR 2020 Workshop on Integration of Deep Neural Models and Differential Equations, 2020.
>
> [2] Zongyi Li, Nikola Kovachki, Kamyar Azizzadenesheli, Burigede Liu, Andrew Stuart, Kaushik Bhattacharya, and Anima Anandkumar. Multipole graph neural operator for parametric partial differential equations. Advances in Neural Information Processing Systems, 33:6755–6766, 2020a
>
> [3] Zongyi Li, Nikola Borislavov Kovachki, Kamyar Azizzadenesheli, Kaushik Bhattacharya, Andrew Stuart, Anima Anandkumar, et al. Fourier neural operator for parametric partial differential equations. In International Conference on Learning Representations, 2020b.
>
> [4] Alasdair Tran, Alexander Mathews, Lexing Xie, and Cheng Soon Ong. Factorized fourier neural operators. In The Eleventh International Conference on Learning Representations, 2022.